# SARS-CoV-2 neutralizing antibodies: Longevity, breadth, and evasion by emerging viral variants

**Fiona Tea**[1], **Alberto Ospina Stella**[2], **Anupriya Aggarwal**[2], **David Ross Darley**[3,4], **Deepti Pilli**[1], **Daniele Vitale**[5], **Vera Merheb**[1], **Fiona X. Z. Lee**[1], **Philip Cunningham**[6], **Gregory J. Walker**[7], **Christina Fichter**[2], **David A. Brown**[5,7], **William D. Rawlinson**[7,8,9], **Sonia R. Isaacs**[7], **Vennila Mathivanan**[2], **Markus Hoffmann**[10,11], **Stefan Pöhlman**[10,11], **Ohan Mazigi**[4,12], **Daniel Christ**[4,12], **Dominic E. Dwyer**[7,13,14], **Rebecca J. Rockett**[13,14], **Vitali Sintchenko**[7,13,14,15], **Veronica C. Hoad**[16], **David O. Irving**[16,17], **Gregory J. Dore**[2,3], **Iain B. Gosbell**[16,18], **Anthony D. Kelleher**[2‡], **Gail V. Matthews**[2,3‡], **Fabienne Brilot**[1,14,19,20‡]*, **Stuart G. Turville**[2‡]

1 Brain Autoimmunity Group, Kids Neuroscience Centre, Kids Research at the Children's Hospital at Westmead, Sydney, New South Wales, Australia, 2 The Kirby Institute, The University of New South Wales, Sydney, New South Wales, Australia, 3 St Vincent's Hospital, Sydney, New South Wales, Australia, 4 School of Medicine, St Vincent's Clinical School, The University of New South Wales, Sydney, New South Wales, Australia, 5 Westmead Institute for Medical Research, Sydney, New South Wales, Australia, 6 St Vincent's Applied Medical Research, Sydney, New South Wales, Australia, 7 New South Wales Health Pathology, Sydney, Australia, 8 School of Medical Sciences, Biotechnology and Biomolecular Sciences and School of Women's and Children's Health, The University of New South Wales Sydney, New South Wales, Australia, 9 Serology and Virology Division (SAViD), NSW HP SEALS, Randwick, Australia, 10 Infection Biology Unit, German Primate Center, Göttingen, Germany, 11 Faculty of Biology and Psychology, Georg-August-University Göttingen, Göttingen, Germany, 12 Garvan Institute of Medical Research, Sydney, New South Wales, Australia, 13 Centre for Infectious Diseases & Microbiology, Public Health, New South Wales Health Pathology, Institute of Clinical Pathology & Medical Research (ICPMR), Westmead, Sydney, New South Wales, Australia, 14 Marie Bashir Institute for Biosecurity, Faculty of Medicine and Health, The University of Sydney, Sydney, New South Wales, Australia, 15 Faculty of Medicine and Health, The University of Sydney, Sydney, New South Wales, Australia, 16 Australian Red Cross Lifeblood, Melbourne, Victoria, Australia, 17 Faculty of Health, University of Technology, Sydney, New South Wales, Australia, 18 School of Medicine, Western Sydney University, Sydney, New South Wales, Australia, 19 School of Medical Sciences, Faculty of Medicine and Health, The University of Sydney, Sydney, New South Wales, Australia, 20 Brain and Mind Centre, The University of Sydney, Sydney, New South Wales, Australia

☯ These authors contributed equally to this work.
‡ SGT, FB, ADK, and GVM also contributed equally to this work.
* Fabienne.brilot@sydney.edu.au

**Data Availability Statement:** Data cannot be shared publicly because of ethical considerations. Data are available from St Vincent Hospital and Lifeblood Institutional Data Access / Ethics Committee (contact via FB) for researchers who meet the criteria for access to confidential data.

## Abstract

The Severe Acute Respiratory Syndrome Coronavirus 2 (SARS-CoV-2) antibody neutralization response and its evasion by emerging viral variants and variant of concern (VOC) are unknown, but critical to understand reinfection risk and breakthrough infection following vaccination. Antibody immunoreactivity against SARS-CoV-2 antigens and Spike variants, inhibition of Spike-driven virus–cell fusion, and infectious SARS-CoV-2 neutralization were characterized in 807 serial samples from 233 reverse transcription polymerase chain reaction (RT-PCR)–confirmed Coronavirus Disease 2019 (COVID-19) individuals with detailed demographics and followed up to 7 months. A broad and sustained polyantigenic immunoreactivity against SARS-CoV-2 Spike, Membrane, and Nucleocapsid proteins, along with high viral neutralization, was associated with COVID-19 severity. A subgroup of "high

Material transfer should be obtained through a Material Transfer Agreement.

**Funding:** This work was supported by Snow Medical (Australia, FB), The University of New South Wales Rapid Response grant (Australia, ADK), the University of Sydney Research Excellence Initiative grant (Australia, FB), the Medical Research Future Fund COVID-19 grant (MRFF2005760, SGT), Medical Research Future Fund Antiviral Development Call grant (DC), Medical Research Future Fund COVID-19 grant (MRFF2001684, ADK), the New South Wales Health COVID-19 Research Grants Round 2 (FB), and the Australian Governments for the provision of blood, blood products, and services for the Australian community (Australian Red Cross Lifeblood, IBG). The funders had no role in study design, data collection and analysis, decision to publish, or preparation of the manuscript.

**Competing interests:** I have read the journal's policy and the authors of this manuscript have the following competing interests: FB has received honoraria from Biogen Idec and Merck Serono as invited speaker. The other authors have declared that no competing interests exist.

**Abbreviations:** ΔMFI, delta median fluorescence intensity; ADAPT, Adapting to Pandemic Threats; COVID-19, Coronavirus Disease 2019; GLM, general linear model; HSD, honestly significant difference; LIFE, Australian Red Cross Lifeblood cohort; NIBSC, National Institute for Biological Standards and Control; OE-PCR, overlap extension polymerase chain reaction; RBD, receptor-binding domain; RECOVERY, Randomized Evaluation of COVID-19 Therapy; RT-PCR, reverse transcription polymerase chain reaction; SARS-CoV-2, Severe Acute Respiratory Syndrome Coronavirus 2; VOC, variant of concern; WHO, World Health Organization.

responders" maintained high neutralizing responses over time, representing ideal convalescent plasma donors. Antibodies generated against SARS-CoV-2 during the first COVID-19 wave had reduced immunoreactivity and neutralization potency to emerging Spike variants and VOC. Accurate monitoring of SARS-CoV-2 antibody responses would be essential for selection of optimal responders and vaccine monitoring and design.

## Introduction

Control of the Severe Acute Respiratory Syndrome Coronavirus 2 (SARS-CoV-2) pandemic relies on population resistance to infection due to a postinfection and vaccination-induced immunity. Current questions relate to the level, breadth, and longevity of generated immunity and whether mutation of the virus will compromise immunity. Previous studies reported varying results in longitudinal changes of the virus-specific antibody response. Some detected stable antibody titers 4 to 6 months after diagnosis [1,2], while others reported waning of the antibody response 2 to 3 months after infection [3,4]. Differences in assay sensitivity and antigen targets may account for these discrepancies, with Spike and Nucleocapsid being the main antigens investigated. Immunoreactivity to other abundant antigens, such as Membrane or Envelope, is unknown.

Neutralization of SARS-CoV-2 has been reported for antibodies that bind to Spike, a large homotrimeric glycoprotein studded across the viral surface [5,6], whereas Membrane and Envelope proteins, although exposed on the viral surface, remain to be identified as neutralizing antibody targets. Rapid development of neutralizing antibody response to Spike correlates with viral immunity, and individuals who seroconvert may develop a lasting neutralization response [7].

The SARS-CoV-2 virus has accumulated many polymorphisms across its genome, especially within the Spike gene [8]. Shortly after the introduction of SARS-CoV-2 into the human population, many early and dominant amino acid polymorphisms were associated with viral entry fitness, such as D614G [9,10]. However, the pressure of the neutralizing antibody response might select for escape mutations in Spike that limit postinfectious immunity or vaccine protection [11]. One example is the S477N/D614G Spike variant, which appeared in Australia during July and August 2020, was traced to a single event from Australian hotel quarantine [12], and represented greater than 58% in Oceania [13]. More recently, several SARS-CoV-2 variants of concern (VOCs) appeared primarily in the United Kingdom (B.1.1.7), South Africa (B.1.351), and Brazil (B.1.1.28.1 and B.1.1.28.2, recently renamed P1 and P2, respectively), and VOC B.1.1.7 is currently becoming the most dominant worldwide [14].

Using the lessons learned from research of other viral pathogens and neuroimmunological autoantibodies [15,16], we have developed a suite of novel high-content assays that sensitively assess antibody responses against the native oligomeric structure of Spike and its emerging variants [17]. To measure the neutralizing capacity, we have also developed a Biosafety Level 2 surrogate Spike-driven virus–cell fusion assay that has been cross-validated with a novel high-content, machine-scored, Biosafety Level 3 authentic SARS-CoV-2 neutralization assay.

Herein, we characterize the longevity, polyantigenic breadth, and neutralization capacity of the SARS-CoV-2 antibody response in individuals and their responses to globally emerging SARS-CoV-2 variants. Using 2 longitudinal SARS-CoV-2 community- and hospital-based Australian cohorts representative of the broad spectrum of disease severity at acute infection,

we showed that the polyantigenic and neutralizing responses to SARS-CoV-2 are sustained, associated with Coronavirus Disease 2019 (COVID-19) severity, and are evaded by emerging viral variants.

This work provides a community snapshot of humoral immunity in those recovering from infection and sheds light on important considerations for vaccine design and selection of donors for convalescent plasma therapy. Additionally, the modular assays used herein can be adapted for novel viral pathogens to respond rapidly to emerging pathogens.

## Results

### SARS-CoV-2 antibody responses are sustained for up to 7 months postinfection and are focused on Spike

SARS-CoV-2 antibodies were assessed in reverse transcription polymerase chain reaction (RT-PCR)–confirmed COVID-19 convalescent adults in 2 Australian cohorts: ADAPT, a hospital-based cohort of patients recruited during the first and second wave of infection in Australia ($n$ = 83 and $n$ = 17), and LIFE, a cohort of plasma donors ($n$ = 159) (Table 1, Fig 1A). Antibody immunoreactivity to SARS-CoV-2 antigens, inhibition of virus–cell fusion, live SARS-CoV-2 neutralization, and immunoreactivity to Spike emerging variants were assessed, and antibody features were compared with demographic data (Fig 1A). At first date of collection postinfection, 96% (81/83 ADAPT, median 71 days, mean 74 days, post first PCR positivity) and 98% (152/159 LIFE, median 59 days, mean 61 days) of infected patients were Spike IgG positive, and 81% (66/83 ADAPT) and 91% (139/152 LIFE) were Spike IgM positive (Table 2). A broad range of Spike IgG levels was observed. No differences in Spike IgG and IgM levels were observed between females and males, but higher IgG and IgM levels were associated with older age ($P < 0.0001$) (S1 Fig). Detection of convalescent positive serostatus was more sensitive when Spike IgG were detected by live-cell flow cytometry compared to Nucleocapsid IgG or Spike IgG using commercial assays (Table 2).

The longevity of antibody responses was assessed in 807 Spike IgG-positive serial samples from 233 individuals ($n$ = 162 ADAPT and $n$ = 645 LIFE), spanning up to 205 days post-PCR positivity (Fig 1A, Table 1). There was a range of Spike IgG titers at first collection date, and among all Spike IgG-positive individuals, no individual seroreverted, even up to 205 days post-PCR positivity. The majority of ADAPT patients had stable IgG responses (85%), whereas most LIFE donors exhibited decreased IgG over time (59%), where a decrease was defined as >30% change from first collected sample (Fig 1B) [15]. A 2-phase decay in those with decreasing responses characterized by an initial high rate of decay followed by stabilization, and the breakpoint between the 2 phases was estimated at 85 days post-PCR positivity (Fig 1B). The level at which Spike IgG stabilized was dependent on intial antibody response. High Spike IgG levels decayed to mid-level reactivity and mid-low level reactivity to low level (Fig 1B). In Spike IgM-positive patients, the majority had decreased IgM levels over time (68% ADAPT; 84% LIFE), in which levels initially decreased and then stabilized at lower levels, but did not serorevert up to 205 days (Fig 1C). Only 5 ADAPT (6%) and 14 LIFE (9%) individuals seroreverted for Spike IgM at median 146 days post-PCR positivity (Fig 1C). The breakpoint between the 2 phases of IgM decay was at 93 days post-PCR positivity.

The polyantigenic breadth of Spike IgG-positive individuals against the virus was examined by detecting IgG targeting the SARS-CoV-2 Membrane, Envelope, and Nucleocapsid proteins (Table 2). A total of 54% (45/83 ADAPT) and 57% (87/152 LIFE) individuals harbored IgG targeting the SARS-CoV-2 Membrane protein, whereas 78% had antibody targeting the Nucleocapsid protein (65/83 ADAPT, 118/152 LIFE) (Fig 1D and 1E). Antibody titers toward the Membrane protein remained stable over the period of observation in most individuals (91%,

**Table 1. Demographics of the convalescent SARS-CoV-2 ADAPT and LIFE cohorts.**

| | ADAPT | | LIFE |
|---|---|---|---|
| **Timeline**[*] | **First wave** | **Second wave** | **First wave** |
| RT-PCR–confirmed patients n | 83 | 17[**] | 159 |
| Serial samples n | 166 | - | 645 |
| Gender Male:female (ratio) | 35:48 (0.7) | 6:11 (0.5) | 72:79 (0.9) |
| Age at RT-PCR positivity Median years (IQR, min, max) | 48 (35–59, 20, 79) | 44 (34–64) | 51 (30–63, 19, 78) |
| Days after RT-PCR positivity at first sample collection Median days (IQR, min, max) Mean days (SD) | 71 (64–86, 36, 122) 74 (16) | 31 (26–39, 21, 47) 33 (8) | 59 (52–67, 33, 100) 61 (12) |
| Days after RT-PCR positivity at latest sample collection Median days (IQR, min, max) Mean days (SD) | 118 (115–132, 114, 139) 123 (12) | n/a | 95 (77–126, 55, 205) 127 (39) |
| **Disease severity at acute infection**[***] | | | |
| Nonhospitalized n (% total) | 73 (88) | 17 (100) | 145 (95)[#] |
| Mild n (%) | 31 (42) | 8 (47) | n/a |
| Moderate n (%) | 42 (58) | 9 (53) | n/a |
| Hospitalized n (%) | 10 (12) | 0 (0) | 7 (5)[‡] |
| Admitted to ICU n (%) | 3 (30) | 0 (0) | - |

[*]Timeline according to Fig 1A.

[**]n = 8 infected and PCR positive for S477N/D614G and n = 9 infected with and PCR positive for S477N/D614G/V1068F.

[***]Non-hospitalized Mild: community managed with minor, mostly upper respiratory tract viral symptoms including sore throat, rhinorrhoea, headache, and anosmia/ageusia. Nonhospitalized Moderate: community managed with fever/chills and 1 or ≥ 2 of the following organ-localizing symptoms; cough, haemoptysis, shortness of breath, chest pain, nausea/vomiting, diarrhea, or altered consciousness/confusion. Hospitalized: inpatient ward care. Hospitalized and admitted to ICU: care in the ICU for acute respiratory distress syndrome.

[#]No data on disease severity or symptoms were collected in LIFE.

[‡]Information on hospitalization was self-reported in LIFE.

ICU, intensive care unit; RT-PCR, reverse transcription polymerase chain reaction; SARS-CoV-2, Severe Acute Respiratory Syndrome Coronavirus 2.

41/45 ADAPT; 95%, 83/87, LIFE), whereas responses toward the Nucleocapsid protein differed between ADAPT and LIFE and were reminiscent of the Spike IgG response, i.e., mostly stable in ADAPT and mostly decreased in LIFE over time (Fig 1D and 1E). Across both cohorts, reactivity to the Envelope protein was very limited with only 2 ADAPT patients (2%) positive for Envelope IgG (Table 2). Antibody responses to SARS-CoV-2 were highly focused on Spike, followed by the Nucleocapsid and Membrane proteins. Individuals with higher Spike IgG had also high levels of Nucleocapsid and Membrane IgG (S1 Fig).

The overall decay of SARS-CoV-2 antibodies between both cohorts behaved similarly for Spike IgM, but not for Spike IgG, Membrane IgG, and Nucleocapsid IgG, with LIFE donors

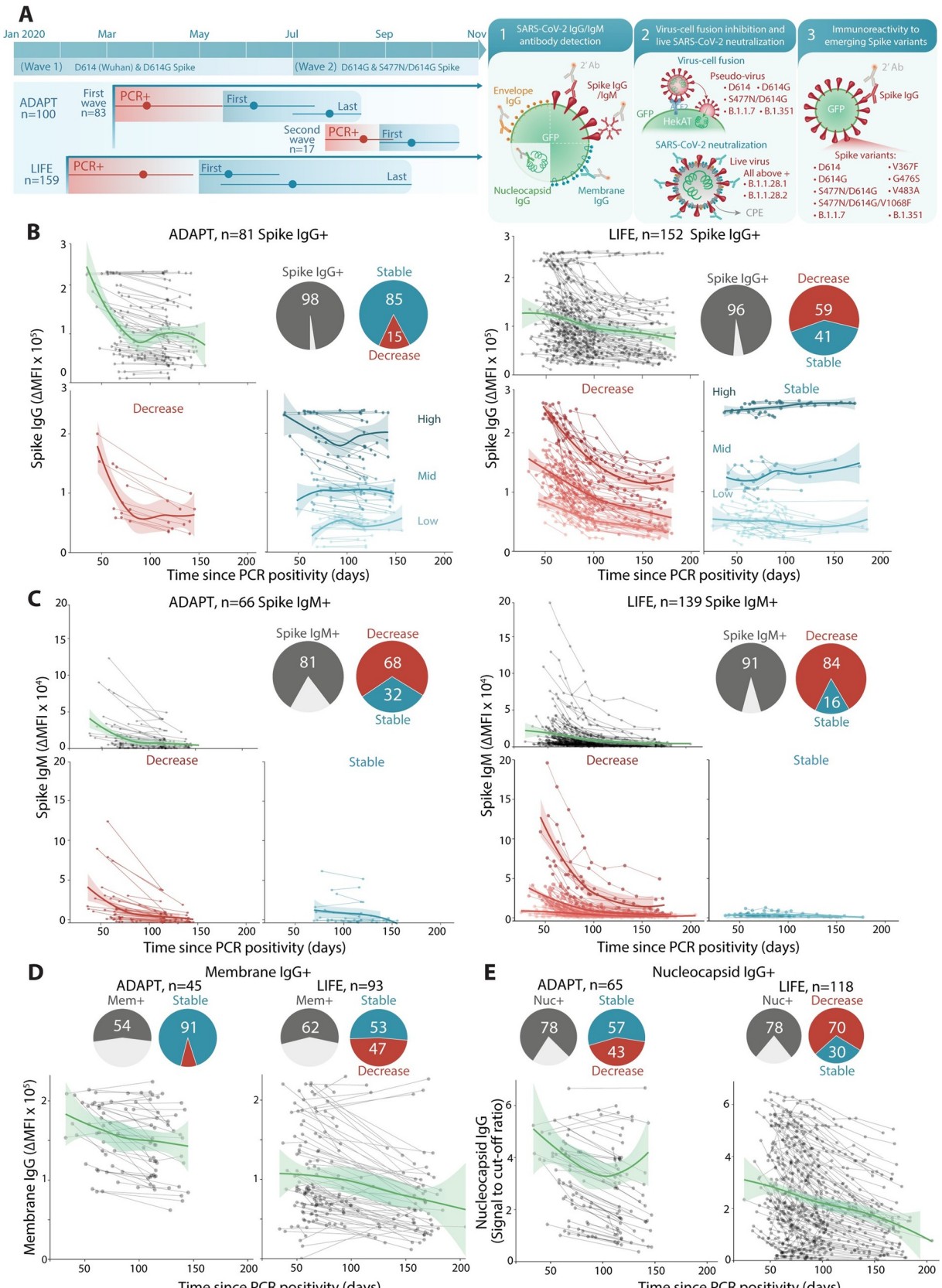

**Fig 1. SARS-CoV-2 antibody responses are sustained and are predominantly focused on Spike. (A)** The first wave of Australian infections were from D614 and D614G Spike and the S477N/D614G Spike variant emerged during the second wave. Convalescent patient sera from ADAPT (first and second waves) and LIFE (first wave) were examined for SARS-CoV-2 antibodies. Mean time and range of PCR positivity (red) and dates of first and last sample collection (blue) are shown. Seropositive patients with at least 3 weeks between first and last samples were examined over time. Summary schematic of the current study that includes examination of (1) patient antibody responses toward various SARS-CoV-2 antigens; (2) functional virus–cell fusion and viral neutralization; and (3) immunoreactivity toward emerging Spike variants and VOC (S1 Table). **(B)** 96%–98% (gray) of patients were Spike IgG+. Most ADAPT patients had stable levels overtime, whereas most of LIFE Spike IgG levels decreased. No patients seroreverted. **(C)** 81%–91% (gray) were Spike IgM+, most had decreasing levels over time, and Spike IgM+ individuals started with and maintained low IgM levels. **(D)** 54%–57% (gray) of sera were Membrane IgG+, and most ADAPT had stable levels, whereas a larger proportion of LIFE had decreasing levels. **(E)** 78% of sera were Nucleocapsid IgG+, most were stable in ADAPT, whereas most decreased in LIFE. Loess curves with 95% confidence intervals are shown. PCR, polymerase chain reaction; SARS-CoV-2, Severe Acute Respiratory Syndrome Coronavirus 2.

exhibiting a higher proportion of decreased profiles (Fig 1). The first collected sample in ADAPT started later post-PCR positivity, and the time duration between paired samples was shorter than for LIFE samples; therefore, some ADAPT patients may have been captured during the second, more stable, phase (Fig 1, Table 1). Furthermore, few ADAPT patients underwent plasmapheresis, whereas all LIFE donors underwent plasmapheresis as part of convalescent plasma donations (median 6 donations, IQR 3 to 9, max 14). However, donors

**Table 2. Comparison of the sensitivity of SARS-CoV-2 antibody detection assays.**

| | ADAPT (n = 166 samples) | | LIFE | | | |
| | | | All samples (n = 645) | | First and last samples (n = 302*) | |
| | Positive samples, n (%) | Sensitivity % (95% CI) | Positive samples, n (%) | Sensitivity% (95% CI) | Positive samples, n (%) | Sensitivity% (95% CI) |
|---|---|---|---|---|---|---|
| **Spike IgG** | | | | | | |
| Flow cytometry assay | 162 (98) | 98 (94–99) | 645 | n/a | 302 | n/a |
| EUROIMMUN | 121 (73) | 73 (65–80) | n/a | n/a | n/a | n/a |
| **Spike IgM** | | | | | | |
| Flow cytometry assay | 127 (77) | 76** (70–83) | 608 (94) | 94**, ^^ (92–96) | 276 (91) | 91**, ^^ (87–94) |
| **S1/S2 Spike IgG** | | | | | | |
| DiaSorin LIAISON SARS-CoV-2 S1/S2 IgG assay | 134 (81) | 81 (74–86) | n/a | n/a | n/a | n/a |
| **Nucleocapsid IgG** | | | | | | |
| Abbott Architect SARS-CoV-2 assay | 116 (70) | 70 (62–77) | 472 (73) | 73 (70–77) | 222 (74) | 74 (68–78) |
| EUROIMMUN | 121 (73) | 73 (65–80) | n/a | n/a | n/a | n/a |
| **Nucleocapsid/Spike IgG***** | n/a | n/a | 577 (89) | 89 (87–92) | 271 (90) | 90 (86–93) |
| **Membrane IgG** | | | | | | |
| Flow cytometry assay | 87 (52) | 52 (45–60) | n/a | n/a | 173 (57) | 57 (51–63) |
| **Envelope IgG** | 4 (2) | 2 (0.8–6) | 0 | 0 | 0 | 0 |

*Only first and last samples of LIFE cohort were tested for Membrane IgG.

**Sensitivity is influenced by IgM seroreversion in 5 ADAPT and 14 LIFE donors.

***Positivity determined using a 2-step clinical diagnostic testing with the Abbott Architect SARS-CoV-2 Nucleocapsid IgG assay, followed by the EUROIMMUN Spike IgG assay.

^^ Please refer to the materials and methods for time of collection as it affects the IgM serostatus

SARS-CoV-2, Severe Acute Respiratory Syndrome Coronavirus 2.

with more than 10 donations ($n$ = 30) had decay profiles similar to the whole cohort, in which donors stabilized at mid-low level, and none of these highly recurrent donors became seronegative (S2 Fig).

## Neutralization of SARS-CoV-2 is correlated with Spike antibody levels and is maintained over time

The neutralization capacity of these individual responses was assessed on a Spike-driven virus–cell fusion assay and a whole-virus neutralization assay (Fig 1A). Most sera were capable of inhibiting virus–cell fusion (82%, 68/83 ADAPT; 68%, 104/152 LIFE) and mediating viral neutralization (88%, 73/83 ADAPT; 94%, 143/152 LIFE) (Fig 2A, Table 2). In both cohorts, the virus–cell fusion assay was more stringent than the SARS-CoV-2 neutralization assay as a proportion of individual sera with lower titers in the SARS-CoV-2 neutralization assay were negative in the virus–cell fusion assay (7%, 6/83 ADAPT; 27%, 41/152 LIFE), and most individuals had higher titers in the neutralization assay (Fig 2A). To understand the discrepancy between both viral assays, live SARS-CoV-2 viral particles were enumerated and directly compared to Spike-pseudotyped lentiviral particles. Cell-permeable RNA-specific staining of live virions detected viral particles that were Nucleocapsid positive (Fig 2B). The particle to transduction ratios from the fusion assay were $1.03 \times 10^5$, consistent with the low specific infectivity of lentiviruses such as HIV-1[18]. In contrast, the SARS-CoV-2 particle to infectivity ranged from 58 (HekAT14) to 578 (VeroE6), consistent with the ratio reported for influenza virus [19]. However, the absolute viral particle number was 74-fold higher in Spike-pseudotyped particle preparation ($1.64 \times 10^8$ particles per ml) compared to authentic SARS-CoV-2 ($2.22 \times 10^6$ particles per ml). Thus, the specific infectivity of SARS-CoV-2 was higher than that of Spike-expressing lentiviral particles, which may account for the higher sensitivity of the SARS-CoV-2–based neutralization assay.

Most ADAPT patients had stable virus–cell fusion inhibition (99%) and neutralization (89%) titers over time (Fig 2C and 2D). Most of LIFE donors had decreased virus–cell fusion inhibition (82%) and neutralization (56%) capacity over time, and the majority exhibited a single-phase decay in both assays (Fig 2C and 2D). The greater number of samples per LIFE donor enabled finer characterization of the decay profile in 34 donors in the virus–cell fusion and 44 donors in the neutralization assay. Most donors had a single-phase decay, while a 2-phase decay was observed in those with >1:320 titers at first collection. These rapidly dropped and then stabilized over time at 1:80 to 1:160 (28% and 25% of LIFE donors in the virus–cell fusion and neutralization assays, respectively). Individuals with 2-phase decay had much higher starting titers than individuals with a single-phase decay (Fig 2D). In the neutralization assay, LIFE donors with decreased profile had a similar median follow-up as the stable profile (approximately 63 days and 56 days, respectively). These results were similar when only positive fusion and neutralization were included, i.e., titers above 1:40. In both cohorts, the neutralization and fusion profiles were similar to the Spike IgG profiles, in which ADAPT had more stable responses than LIFE. Indeed, Spike IgG and IgM titers were strongly correlated with virus–cell fusion inhibition and SARS-CoV-2 neutralization (Fig 2E).

## A broad antigenic repertoire and high neutralization capacity against SARS-CoV-2 is associated with COVID-19 severity

Approximately half of individuals (55% ADAPT and 49% LIFE) had broad polyantigenic immunoreactivity as defined by IgG responses against each of SARS-CoV-2 Spike, Membrane, and Nucleocapsid proteins (Fig 3A). Interestingly, the 2 individuals positive for Envelope IgG also had antibodies against all other SARS-CoV-2 proteins. Around a third of individuals

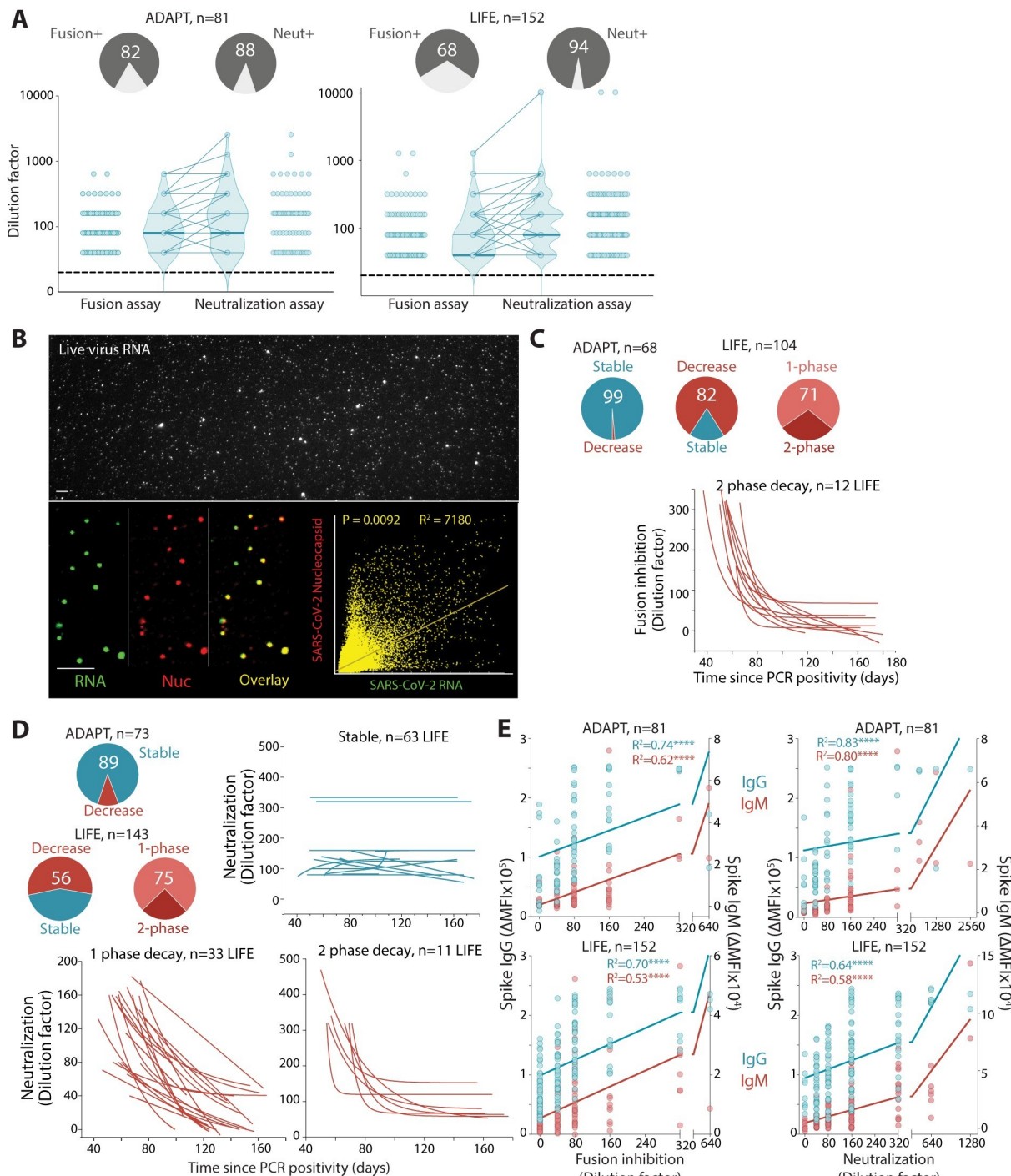

**Fig 2. Viral neutralization and inhibition of viral–cell fusion are strongly correlated with Spike antibody titers and sustained overtime.**
**(A)** 68%–82% of convalescent sera inhibited virus–cell fusion, whereas 88%–94% sera neutralized live authentic SARS-CoV-2. Neutralization titers were higher than viral–cell fusion titers in ADAPT (ns, not significant) and LIFE ($P < 0.0001$). **(B)** Approximately 75% of virus particles were SARS-CoV-2 Nucleocapsid- and RNA positive (overlay, yellow). **(C)** All but one ADAPT patient had stable responses over time, whereas most LIFE donors (82%) had a decreased virus–cell fusion over time, with the majority (71%) exhibiting a single-phase decay. **(D)** In sera capable of viral neutralization, most ADAPT sera were stable (89%), whereas most LIFE sera (56%) had a decreased score over time, with the majority (75%) exhibiting a single-phase decay. Serum curves unable to be fitted were classified as undetermined. **(E)** Spike IgG and IgM levels were correlated with inhibition of virus–cell fusion and neutralization scores. $R^2$ values are shown, and **** indicates significance ($P < 0.0001$). SARS-CoV-2, Severe Acute Respiratory Syndrome Coronavirus 2.

exhibited antibodies against only 2 proteins (27% and 30%, Nucleocapsid and Spike; 2% and 9% toward Membrane and Spike in ADAPT and LIFE respectively), and a smaller proportion had responses against Spike alone (12 and 17%) (Fig 3A). Polyantigenic immunoreactivity did not change overtime in most individuals (82%, 15/81 ADAPT, 83%, 41/152 LIFE). No individual developed IgG to new antigens at any point of follow-up, but, instead, lost immunoreactivity to one antigen, either Nucleocapsid or Membrane.

Patients had broader responses across the spectrum of severity in ADAPT (Fig 3B). ADAPT and LIFE hospitalized patients with more severe symptoms were more likely to exhibit a broader antibody response to SARS-CoV-2, i.e., polyreactive toward the 3 antigens (Fig 3B). Interestingly, 2 of 7 hospitalized LIFE patients who had a short 24-hour hospitalization harbored non-broad responses, and Spike-only responses were exclusively observed in nonhospitalized, mild, and moderate individuals (Fig 3B). Higher IgG titers against Membrane and Nucleocapsid proteins were also associated with disease severity in both cohorts (Fig 3C). Patients with broader SARS-CoV-2 responses and higher disease severity had greater viral neutralization and virus–cell fusion inhibition (Fig 3D and 3E). This polyreactive, high severity subgroup was populated almost exclusively by older males (Fig 3F). Similarly, higher neutralization and virus–cell fusion inhibition titers were more enriched in older males with moderate disease and who were hospitalized (Fig 3G and 3H).

## High responders with strong and broad SARS-CoV-2 antibody responses are rare and could be ideal plasma donors

A small subgroup of individuals were "high responders" characterized by high Spike IgG, Spike IgM positive, broad polyantigenic immunoreactivity (binding to Nucleocapsid, Spike, and Membrane), virus–cell fusion inhibition (>1:160), and neutralization (>1:320). They maintained this high response over time (n = 14, 17% ADAPT, n = 19, 12% LIFE). High responders were more likely to be male, hospitalized, and were of older age (Fig 4A). Further characterization was performed on a series of increasingly permissive cell lines: VeroE6, HekAT14, HekAT10, and HekAT24 (Fig 4B, S3 Fig). Low, i.e., non-high responders, and high responders sera neutralized live SARS-CoV-2 in VeroE6, HekAT14, and HekAT10 cell lines, whereas limited neutralization was observed in the hyperpermissive HekAT24 cell line (Fig 4B). Using the HekAT24 cell line, 2 elite responders were identified in LIFE (Fig 4C), with high Spike IgG and IgM levels, and neutralization titers 30- to 4-fold greater than other individuals (Fig 4D). Interestingly, elite responders had the highest detectable IgM levels, and early IgM decay coincided with a decrease in neutralization titers, whereas Spike IgG remained stable overtime (Fig 4E). This association between decreased IgM and neutralization titers was observed in approximately 10% of individuals in both cohorts.

## Spike IgG antibody binding and neutralizing capacity are dependent on Spike mutations in emerging new variants

Numerous Spike polymorphisms have evolved over the course of the pandemic [11], with the most attention given to the transmission fitness gain variants, such as D614G [10,11], and recent emerging variants from the United Kingdom, South Africa, and Brazil [14]. To test the breadth of the antibody response, Spike IgG immunoreactivity to several Spike variants implicated in the receptor-binding domain (RBD) and S1 was assessed (Fig 1A). Expression of all Spike variants was similar across each transfected cell line used in the flow cytometry antibody assays (S4A and S4B Fig). Compared to the Wuhan-1 D614 variant, most patients had similar binding and were able to recognize the Spike RBD variants G476F, V483A, and V367S (Fig 5A). However, across both cohorts, there was an overall reduced binding to D614G, a

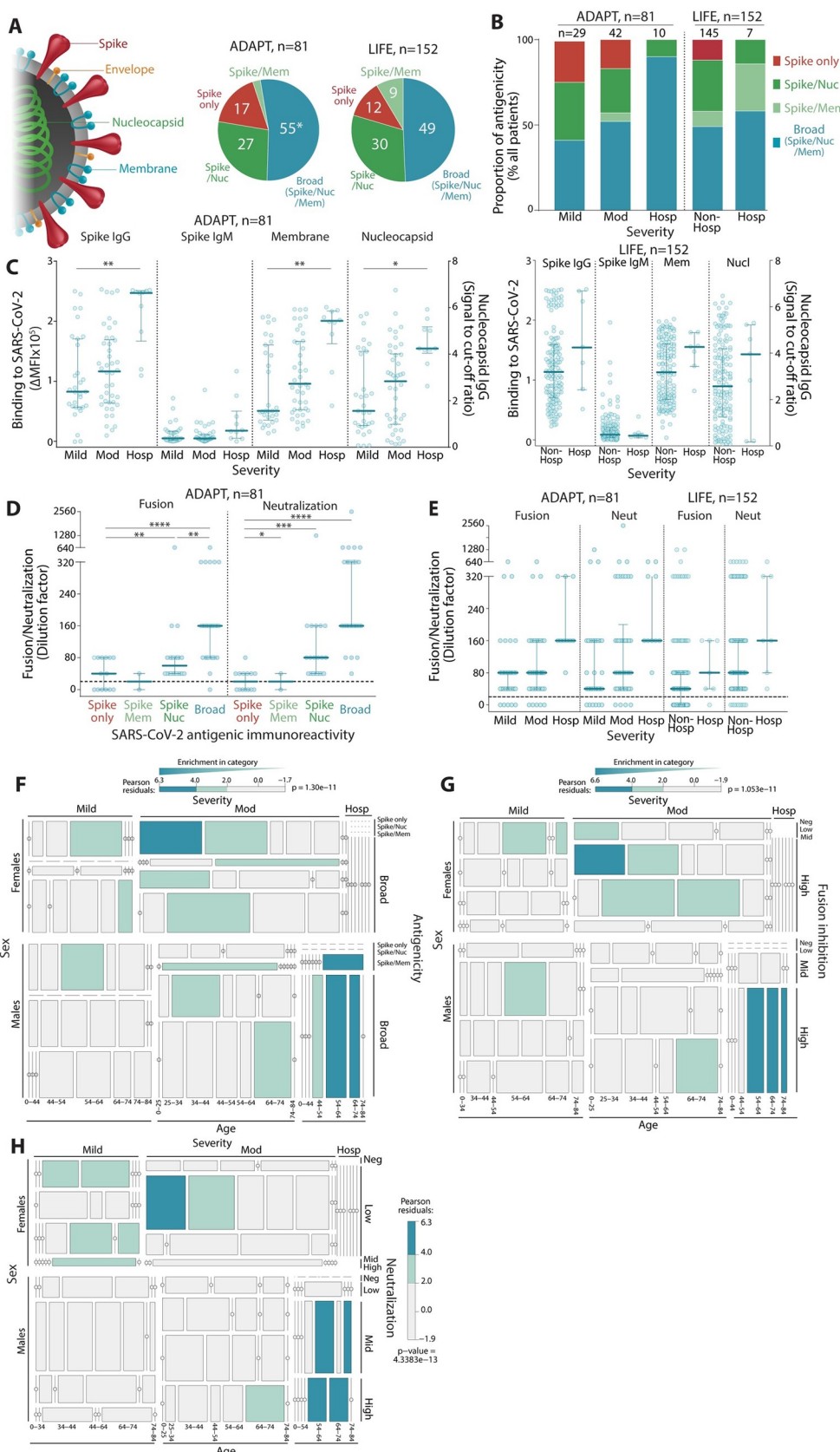

**Fig 3. The antibody responses of patients with more severe COVID-19 disease have broader SARS-CoV-2 polyantigenicity. (A)** Approximately half of patients (49%–55%) had broad SARS-CoV-2 antibodies (blue, * includes $n = 2$ patient seropositive for Envelope IgG). Some had responses to 2 antigens (light and dark green), and a few reacted to Spike only (red). **(B)** Hospitalized patients were more likely to have broad SARS-CoV-2 polyantigenic immunoreactivity, whereas patients with only Spike reactivity exhibited mild-moderate symptoms. **(C)** Hospitalized patients exhibited higher Spike IgG, IgM, Membrane IgG, and Nucleocapsid IgG levels. High virus–cell fusion inhibition and neutralization titers were observed in patients with broad polyantigenic immunoreactivity **(D)** and in hospitalized patients **(E)**. Older males were more likely to present with broader polyantigenic immunoreactivity **(F)**, higher virus–cell fusion inhibition **(G)**, and neutralization scores **(H)**. Younger females were more enriched in mild to moderate disease severity, with narrow antigenicity **(F)**, and lower virus–cell fusion inhibition **(G)** and neutralization scores **(H)**. *: $P < 0.05$, **: $P < 0.01$, ***: $P < 0.001$, ****: $P < 0.0001$, *****: $P < 0.00001$. Not significant if significance is not indicated. $P$ values in LIFE hospitalized patients **(C)** were not calculated due to low sample size. COVID-19, Coronavirus Disease 2019; SARS-CoV-2, Severe Acute Respiratory Syndrome Coronavirus 2.

prominent non-RBD S1 variant present during the Australian first wave (Fig 5A). Moreover, 65% of ADAPT and 91% of LIFE individuals, infected from the first worldwide wave, generated antibodies that bound broadly to G476F, V483A, V367S, and D614G Spike, whereas 35% of ADAPT and 9% of LIFE had more restricted Spike recognition; i.e., they recognized G476F, V483A, and V367S but had a decreased binding to D641G (Fig 5A). Immunoreactivity toward all Spike variants was stable overtime in most patients.

Importantly, sera with reduced D614G IgG binding also had lower neutralization and virus–cell fusion inhibition compared to those who recognized D614G Spike (Fig 5B), suggesting implications for blocking infection in patients who cannot induce robust Spike antibody recognition. Furthermore, patients who bound D614G Spike had broad SARS-CoV-2 polyantigenic immunoreactivity, whereas patients who displayed reduced binding to D614G had more limited antigenic recognition, with 36% recognizing Spike only (Fig 5C). In a D614G virus–cell fusion assay, patients who maintained binding to D614G showed enhanced virus–cell fusion inhibition, compared to when parental Wuhan-1 D614 Spike was used (Fig 5D). Individuals with lower IgG binding to D614G, i.e., restricted variant recognition had limited D614G Spike virus–cell fusion inhibition, and most (8/11) were unable to prevent Spike fusion, emphasizing the need to maintain robust binding to Spike variants for efficient viral neutralization. Patients with restricted Spike variant recognition were not distinguished by age and severity, but were more likely to be female (S5 Fig).

Although D614G Spike remains a predominant variant globally, in the second wave of Australian infection between July to September, an isolate with additional polymorphisms, primarily S477N, and in some cases an additional V1068F, was identified (Fig 5E). These variants were not detected during the first Australian wave, which included the original Wuhan-1 D614 or the D614G variant equally (Fig 5E). To assess the antibody binding capacity between original and emerging variants, patients infected by 2 Spike variants, S477N/D614G and S477N/D614G/V1068F, were recruited during the second wave in Australia ($n = 17$, from the ADAPT cohort, Table 1). All ADAPT patients from the first and second wave had detectable IgG against all Spike variants (S4C Fig). Compared to the D614G variant, a strong decrease in immunoreactivity to S477N/D614G and S477N/D614G/V1068F was observed in all ADAPT patients from the second wave, whereas the third mutation within the Spike S2 domain V1068F did not have an additive effect (Fig 5F). This decrease was also observed irrespective of the virus variant that had infected the ADAPT patients (Fig 5F) and was not due to an N-glycosylation of 477N (S6 Fig). Importantly ADAPT patients from the first wave, who had not encountered the new variants, had reduced binding to S477N/D614G and S477N/D614G/V1068F, suggesting a global decrease of immunoreactivity toward both new variants (Fig 5F). To determine the functional implications of this reduced antibody binding, 48 Spike IgG-positive ADAPT patients ($n = 31$ first wave, $n = 17$ second wave) were assessed for S477N/D614G

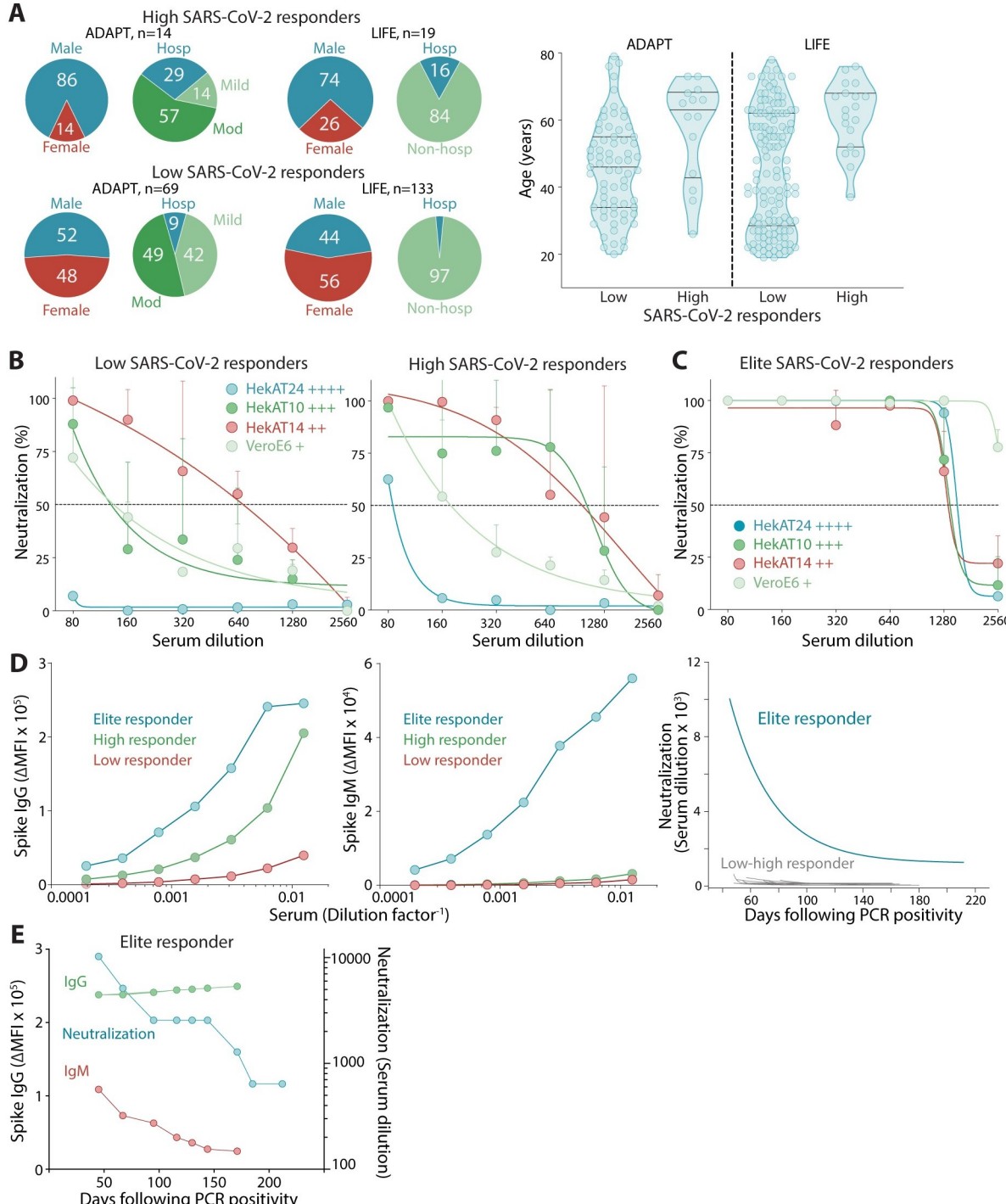

**Fig 4. High and elite responders are discriminated with SARS-CoV-2–permissive cells. (A)** Patients with high and robust SARS-CoV-2 responses were more likely male, hospitalized (left), and of older age (right). **(B)** Low and high responders to SARS-CoV-2 showed limited neutralization in hyperpermissive HekAT24 clonal cells. Permissiveness is indicated by +. **(C)** Only elite responders showed neutralization in HekAT24 cells. **(D)** Serum titration curves from an elite responder (blue) showed IgG and IgM levels greater than low (red) and high (green) responders and incredibly high neutralization titers (≥10,000) that decreased and stabilized at high levels (≥1280). **(E)** The elite donor demonstrated stable high Spike IgG, but the early decrease in viral neutralization was parallel to IgM decline before stabilization (at high titer). PCR, polymerase chain reaction; SARS-CoV-2, Severe Acute Respiratory Syndrome Coronavirus 2.

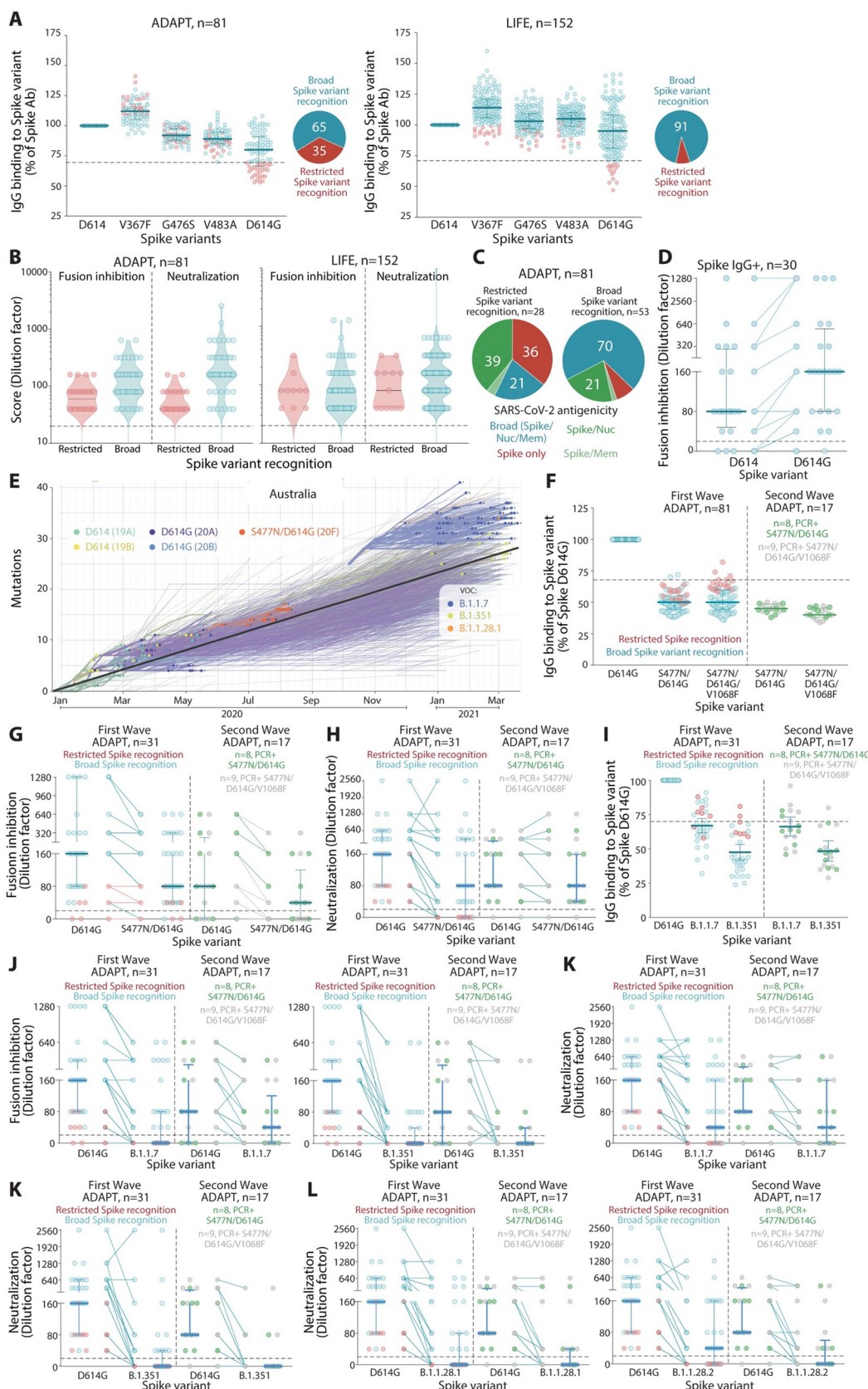

**Fig 5. SARS-CoV-2 antibody responses show evasion by emerging Spike variants. (A)** Most patients had broad recognition of Spike variants (blue), whereas a smaller group had restricted Spike variant recognition and did not have a strong immunoreactivity to D614G Spike (red). Patients with reduced binding to D614G Spike had lower virus–cell fusion (ADAPT $P < 0.01$, LIFE $P < 0.05$) and neutralization scores (ADAPT $P < 0.0001$, LIFE $P < 0.05$) **(B)** and presented with less broad polyantigenic SARS-CoV-2 recognition **(C)**. **(D)** D614G Spike-binding sera had greater inhibition of D614G Spike-pseudotyped virus–cell fusion (ns). **(E)** In Australia, D614G Spike was the predominant variant during the first wave and acquired additional mutations during the second wave (S477N, V1068F). VOCs, with high mutations within Spike, appeared in late December 2021. Pango lineages and Clades are shown in brackets. Graph adapted from Nextstrain [54]. **(F)** All patients had decreased immunoreactivity to S477N/D614G and S477N/D614G/V1068F Spike, while V1068F did not have an additive effect (ns, not significant). **(G)** Patients had reduced virus–cell fusion inhibition (first wave $P < 0.0001$, second wave $P < 0.05$) and neutralization (first wave $P < 0.01$, second wave ns) **(H)** to the S477N/D614G Spike variant compared to D614G. **(I)** Patients had reduced binding to VOC B.1.1.7 (UK) and B.1.351 (SA) Spike, with greater reduction toward B.1.351 (first wave $P < 0.0001$, second wave $P < 0.0001$). Virus–cell fusion inhibition **(J)** and neutralization **(K)** was also reduced against the VOC B.1.1.7 (UK, **(J)** first wave $P < 0.0001$, second wave $P < 0.0001$; **(K)** first wave $P < 0.01$, second wave ns) and B.1.351 (SA), but more so against the VOC B.1.351 (SA, **(J)** first wave $P < 0.0001$, second wave $P < 0.05$; **(K)** first wave $P < 0.00001$, second wave $P < 0.0001$). **(L)** Reduced neutralization was also observed against the authentic VOC B.1.1.28.1 (Brazil) (first wave $P < 0.00001$, second wave $P < 0.01$) and B.1.1.28.2 (Brazil) (first wave $P < 0.00001$, second wave $P < 0.01$). The level of decreased binding **(F, I)**, virus–cell fusion inhibition **(G, J)**, and neutralization **(H, K)** was irrespective of the virus that infected patients during the second wave. PCR, polymerase chain reaction; SARS-CoV-2, Severe Acute Respiratory Syndrome Coronavirus 2; VOC, variant of concern.

virus–cell fusion inhibition and live-virus neutralization. A total of 7 and 1 patient sera were unable to inhibit virus–cell fusion and neutralization, respectively (Fig 5G and 5H). Compared to D614G Spike, most patients had reduced S477N/D614G Spike virus–cell fusion inhibition and neutralization (66%, 27/41, and 74%, 35/47, respectively), and 34% and 19% had similar responses (14/41 and 9/47) (Fig 5G and 5H). Interestingly, patients with reduced S477N/D614G Spike virus–cell fusion inhibition and neutralization had less antibody binding to S477N/D614G Spike than patients with similar fusion inhibition and neutralization, emphasizing the importance of robust Spike binding for potent viral neutralization.

In 2020, B.1.1.7, B.1.351, and B.1.1.28.1 and B.1.1.28.2 appeared at 3 geographical locations. These VOC only represented 7% community transmission in Australia, whereas quarantine detection accounted for 93% in March 2021 (Fig 5E). All ADAPT patients from the first ($n = 31$) and second wave ($n = 17$) were seropositive and had detectable IgG against Spike VOC B.1.1.7 and B.1.351 (S4D Fig). Compared to the D614G variant (parental variant), a strong decrease in immunoreactivity to VOC B.1.1.7 and B.1.351 was observed in all ADAPT patients (Fig 5I). Most patients had reduced Spike virus–cell fusion inhibition and neutralization toward VOC B.1.1.7 and B.1.351, but the proportion of reduced patients and the level of the decrease was greater against VOC B.1.351 (83%, 39/47, and 96%, 45/47, respectively (Fig 5J and 5K). There was also a strong decrease of neutralization to both live VOC B.1.1.28.1 and B.1.1.28.2 (Fig 5L). Interestingly, 4 elite responders were able to maintain neutralization against the 4 live VOC viruses (S7 Fig), suggesting that only a small proportion community infected individuals would harbor immune protection in the event of a reinfection with emerging VOC.

## Discussion

The current study characterizes the breadth, longevity, and neutralizing capacity of SARS-CoV-2 antibody response in 2 Australian cohorts, encompassing a wide range of demographics and disease states, up to 7 months after COVID-19 diagnosis. We show the development of broad and sustained immunoreactivity against SARS-CoV-2 antigens and found high titers of Spike-binding and virus-neutralizing antibodies were associated with COVID-19 severity. A group of high responders were identified with high, broad, and sustained neutralizing responses, who may represent ideal donors for convalescent plasma donations. Most importantly, although most patients seroconverted, antibodies generated after early infection

displayed a significantly reduced antibody binding and neutralization potency to emerging evasive variants. Our data have important implications on hyperimmune therapy, monoclonal antibody treatments, and vaccine development strategies against emerging viral variants.

The longevity of the immune response against SARS-CoV-2 is a fundamental yet currently unresolved question. Like others, we observed a strong correlation between Spike IgG levels and neutralization capacity [20,21]. Although reports on neutralization prevalence and average titers vary widely depending on sampling and detection assay strategies [22,23], our results expand on previous findings by comparing neutralization levels with antigen-specific response over a longer follow-up period with more time points than most previous studies. Early antibody responses against SARS-CoV-2 could not be studied as the first sample was collected beyond 2 months after PCR positivity and is a limitation of the study. The decline in IgG titers and neutralization often stabilized at different levels later into convalescence, addressing whether decreasing IgG levels eventually plateau, this was particularly evident in LIFE whose samples were collected later postinfection and with a longer follow-up period than ADAPT. Spike IgM levels decreased more rapidly than IgG, but were still detectable up to 205 days after diagnosis, much later than previously reported [1,20] and consistent with mathematical modeling of decline of IgM titers in a smaller convalescent cohort [24]. While our results reveal widely different magnitudes of initial responses and a decrease in neutralizing antibodies titers, most patients have detectable Spike IgG and neutralizing responses more than 5 months after diagnosis, suggesting extended humoral protection, even in those with mild manifestations of the disease.

IgG and IgM against conformational Spike antibody assays have been seldom used, and Spike IgM detection has been challenging. Although many serological assays have reported 100% sensitivity at approximately 15 days postinfection [25], prevalence studies, vaccine efficacy, and assessment for convalescent COVID-19 plasma donors may not recruit so early postinfection or postvaccination. In this context, and in future seroprevalence studies, more sensitive antibody assays will be essential despite their comparatively higher costs. Flow cytometry assays are used in clinical diagnostics, mainly in the sensitive and specific detection of neuroimmunological autoantibodies in which antigen conformation and discrimination of seropositive patients from healthy controls are critical [15,16]. Within the follow-up time, the detection of Nucleocapsid and Spike IgG by high capacity commercial assays was less sensitive compared to the flow cytometry assay. Integration of the flow cytometry assay to detect Spike IgG would be valuable to include in the diagnostic pipeline in addition to resource-intensive whole-virus neutralization. Given the sensitivity of the flow cytometry assay, this methodology would be ideally suited toward seroprevalence in populations to reveal the true rates of community infection. More so, due to its modular capacity in assessing Spike IgG binding to emerging variants, the assay could be useful during vaccine studies.

The majority of individuals in both cohorts were treated in the community. COVID-19 severity, from mild to hospitalization, was associated with an antibody immune response against SARS-CoV-2 that was reactive toward an increasing number of SARS-CoV-2 antigens, as recently reported [26]. As our cohorts included only convalescent individuals, the role of broad polyantigenic immunoreactivity in the acute response of hospitalized patients remains unknown. Indeed, reports of patients with absent humoral immune responses have hinted at the role of T cells and innate immune response during the acute disease. Nonetheless, the presence of a broad polyantigenic viral immunoreactivity can be useful to monitor the quality of the antibody response after vaccination.

While the correlation of Spike IgG levels with viral neutralization was strong, high Spike IgM levels were also associated with high viral neutralization in some, especially during the early convalescent days. A lack of somatic mutations was observed in hundreds of cloned

neutralizing human antibodies from convalescent patients [27]. In addition, many antibody precursor sequences were observed in naive B cells from pre-pandemic patient samples, highlighting the importance of preexisting germline antibody sequences in the neutralization response. The lack of somatic mutations observed in IgG may be consistent with IgM being potent in a neutralization response as both isotypes could have similar affinity binding sites for Spike, but with multiple binding sites per molecule on IgM, the avidity for Spike would be higher.

Full virus neutralization and prevention of virus–cell fusion were associated. While many assays aim to assess neutralization surrogates outside of Biosafety Level 3 laboratories, key differences were observed between Spike-driven virus–cell fusion and the authentic SARS-CoV-2 assay. In our study, the particle to transduction ratio in the virus–cell fusion assay was much higher than the SARS-CoV-2 neutralization assay. This is consistent with the respective infectivity of HIV-1 compared to respiratory viruses such as SARS, non-SARS coronaviruses, and influenza viruses [18,19]. The virus–cell fusion assay involves a single round of infection, whereas the full virus in the neutralization assay is replication competent and undergoes multiple rounds of replication over a 3-day culture. Therefore, the spread of the virus must be considered alongside the capacity of antibodies to inhibit the initial single particle entry and blocking of the virus spread between cells. Although the pseudotyping fusion assay had lower sensitivity, and indeed was observed with all variants tested, most individuals across both cohorts and validation controls had titers in this assay with potency ranking similar to full virus neutralization.

Transfusion of convalescent COVID-19 plasma has been proposed as a therapy, with >70 ongoing randomized controlled trials. While initial clinical trials have supported an acceptable safety profile, there is increasing evidence of a lack of therapeutic efficacy in hospitalized patients. One systematic review and meta-analysis and the Randomized Evaluation of COVID-19 Therapy (RECOVERY) trial, the largest randomized, controlled trial involving convalescent plasma to date, recently reported no significant difference in 28-day mortality between hospitalized patients receiving high-titer convalescent plasma and those receiving standard care [28,29]. There remains a question whether high titer units given early or COVID-19 IgG could still provide a benefit for preventing disease progression. Some successful trials used high titer of convalescent COVID-19 plasma delivered within 3 days of hospitalization and showed reduction of disease progression [30–33]. Alternatively, there may be a role in immunosuppressed patients who cannot produce their own antibody response, in particular for patients with agammaglobulinemia [34,35]. Our findings that the immunological response to SARS-CoV-2 is widely heterogeneous, with large variations in SARS-CoV-2 antibodies and neutralization, polyantigenic immunoreactivity, and longitudinal responses, complement these assertions. To take into account the first phase of decay observed during early convalescence, we propose an optimal window for plasmapheresis, up to 100 days post-diagnosis. Furthermore, the occurrence of a small group of individuals, termed "high and elite responders," with high, broadly neutralizing, and sustained SARS-CoV-2 antibody responses over time, may be due to the rapid and lasting generation of memory B cells [36,37]. These patients were likely to be hospitalized older males. While vaccination and in particular boosting of naturally infected invididuals offers a potential alternative source of convalescent plasma, should convalescent plasma be demonstrated to have a role in treatment, these "high and elite" responders could provide a valuable source of convalescent plasma in the future.

A clear advantage of the methodologies used in this study is the capacity of both Biosafety Level 2 pseudotyped fusion and flow cytometry assays to monitor the effects of viral polymorphisms in real time. Indeed with acceleration of global viral spread, we are now observing evolution of viral fitness and/or immune escape across millions of infected people. The initial SARS-CoV-2 fitness gain of D614G appeared very early in the pandemic [9,10] and serves as

the foundation of all current viral variants, including VOC. Zoonosis of a virus is often followed by finer tuning of replication, as observed in the 2014 Ebola outbreak, in which the variant A82V enabled more efficient receptor NPC1 usage [38]. Although D614G is a single polymorphism outside of the RBD, it impacts the RBD positioning and Spike quaternary structure. The release of hydrogen bonds leading to structural changes is proposed to expose Spike to increase ACE2-dependent fusion [39]. RBD exposure in the D614G variant may explain the association with great inhibition of virus–cell fusion in patients who recognized the D614G Spike variant. These results are consistent with recent studies in hamsters [9] and data on protection from the first human vaccine trials in areas where the D614G Spike variant remains prevalent. However, our data also highlighted that a subgroup of patients who displayed limited antibody binding to D614G Spike also had reduced virus neutralization irrespective of the viral variant that had infected them. Given the emergence of the S477N/D614G polymorphism in most patients infected in the Australian second wave and in Europe [12,13], these results highlight concerns regarding fitness gain and immune evasion. Seroconversion was observed in all patients from the first and second wave and good antibody binding to Wuhan-1 D614 and D614G, but there also was a significant decrease in binding, fusion inhibition, and neutralization to S477N/D614G Spike independent of the variant that had infected individuals. While our data on S477N/D614G emphasize a singular polymorphism effect on fitness gain and immune evasion, the 8 and more changes in Spike both within and outside the RBD, but without S477N, in the most recent VOC B.1.1.7, B.1.351, and B.1.1.28.1 underlie that many changes could also result in observations of immune evasion, via less antibody immunoreactivity and a resistance to virus neutralization as shown by our data. The exact polymorphism(s) behind these observations is unknown. Across all VOCs, N501Y is a shared fitness gain that does not contribute to immune evasion [40], whereas E484K in VOC B.1.351, B.1.1.28.1, and B.1.1.28.2, as well as S477N, is associated with the ability to evade monoclonal antibodies [11,41]. Additionally, the appearance of an N-glycosylation site within Spike RBD at position 477N could lead to glycan shielding, as in HIV [42]. This is not consistent with our data. A more likely mechanism would be a conformational change that not only leads to an increase affinity for ACE2 binding, but also enables Spike to render the virus broadly resistant to numerous antibodies [43]. Such polymorphisms are of signficant concern, as they increase viral fitness and also raise the probability of reinfection in convalescent and vaccinated individuals. Unfortunately, reinfections are currently prevalent in Brazil [44], and the results from vaccination efficacy trials in South African both support real-world observations of the challenges we face when variants like B.1.1.28.1, B.1.1.28.2. and B.1.351 enter and spread within communities. Our findings readily imply a need for periodic updates in vaccine design, as for the influenza vaccine [45]. It is highly probable that we will need to address future VOC that provide even greater challenges for current vaccine design, additional spike changes, such as Q498R and/or the merging of S477N with E484K and N501Y, which can lead to further fitness gains. For instance, a Spike RBD with S477N, E484K, Q498K, and N501Y produces a SARS RBD that is 600-fold greater at engaging ACE2 [41,43]. In that context, how such a SARS-CoV-2 infection would proceed and whether current vaccines would provide sufficient protection is presently unknown. Our evidence that S477N/D614G-infected patients have a similar binding to this variant, albeit reduced, compared to first wave patients, may suggest changing the Wuhan-1 D614 Spike to the S477N/D614G variant in vaccine generation may not overcome the resistance of this variant to the neutralizing antibody response. As antibodies against Spike harness the majority of neutralizing activity, selecting the optimal Spike variants in monovalent or multivalent vaccine strategies may be critical.

Our study has important translatable implications to understand the natural history of COVID-19 and reinfection risk and breakthrough infection following vaccination. We have

highlighted that molecular epidemiology and serosurveillance will both be required to detect emerging polymorphisms. Furthermore, sensitive monitoring of antibody binding and neutralization capacity will be paramount in vaccine design strategy and convalescent plasma therapy, and in seroprevalence studies, and this would require involvement of more rapidly adaptive methodologies to characterize the magnitude of the neutralization antibody responses against emerging variants.

## Materials and methods

### Participants

This study investigated 2 cohorts of RT-PCR–confirmed convalescent individuals recruited from February to October 2020 in Australia (Table 1, Fig 1A). The Adapting to Pandemic Threats (ADAPT) cohort included 83 patients diagnosed at a community-based fever clinic whose sera was collected at 2 time points post-PCR positivity during the first wave (March to August, $n$ = 166 samples) [46]. The second wave included sera from 17 patients recruited between July and October. The Australian Red Cross Lifeblood (Lifeblood) cohort (LIFE) included 645 sera samples from 159 donors collected at multiple time points post-PCR positivity (at least 28 days post-recovery) from volunteers presenting to Lifeblood for whole blood or plasma donation. The disease severity of ADAPT patients ranged from mildly symptomatic (mild), community managed (moderate) to critically unwell and hospitalized (hosp), whereas the self-reported disease severity of LIFE donors included community managed (nonhosp) and hospitalized (hosp) (Table 1). A healthy adult noninfected pre-pandemic cohort was collected in Australia and consisted of healthy and noninflammatory neurological disorder donors ($n$ = 24). No reexposure to SARS-CoV-2 and no reinfection was reported. Ethics approval for this study was granted by St Vincent's Hospital (2020/ETH00964) and Lifeblood (30042020) Research Ethics Committees. Written consent was obtained from all ADAPT patients. In LIFE, the donor consent form included a statement that blood donation may be used in research.

### Flow cytometry cell-based assay for detection of SARS-CoV-2 antibodies

A flow cytometry cell-based assay detected patient serum antibodies against SARS-CoV-2 antigens as for neuroimmunological autoantibodies [15,16]. Individual SARS-CoV-2 full-length Spike (Wuhan-1 D614, V367F, G476S, V483A, D614G, S477N/D614G, S477N/D614G/V1068F, B.1.1.7 (United Kingdom), and B.1.351 (South Africa)) [10,11], Membrane, and Envelope proteins were transiently expressed on transfected HEK293 cells. Serum (1:80) was added to live Spike-expressing cells, and Membrane-, and Envelope-expressing cells were treated with 4% paraformaldehyde and 0.2% saponin, followed by AlexaFluor 647-conjugated anti-human IgG (H+L) (Thermo Fisher Scientific) or anti-human IgM (A21249, Thermo Fisher Scientific, USA). Cells were acquired on the LSRII flow cytometer (BD Biosciences, USA). Patients were SARS-CoV-2 antibody positive if their delta median fluorescence intensity ($\Delta$MFI = MFI transfected cells − MFI untransfected cells) was above the positive threshold (mean $\Delta$MFI + 4SD of 24 pre-pandemic controls) in at least 2 of 3 quality-controlled experiments [15]. Furthermore, 24 pediatric pre-pandemic controls and 8 PCR–negative pandemic adults, among which one patient was infected by coronavirus 229E, were also tested,and were all seronegative for Spike IgG. The assay was verified using 10 antibody standards from the National Institute for Biological Standards and Control (NIBSC). These were tested using the standard protocol for this study. These NIBSC standards were distributed during the CS678 protocol for the World Health Organization (WHO) collaborative study to establish the first International Standard for anti-SARS-CoV-2 antibody and Reference Panel, which our

laboratory participated in. Of the 10 NIBSC standards, 8 were positive in the assay either with low, mid, and high levels of Spike IgG, consistent with the confirmed results from the NIBSC. Binding to Spike variants was expressed as a percentage of reduced binding compared to Spike. Data were analyzed using FlowJo 10.4.1 (TreeStar, USA), Excel (Microsoft, USA), and GraphPad Prism (GraphPad Software, USA).

## Commercial SARS-CoV-2 ELISA

Nucleocapsid IgG assay on the ARCHITECT-I (Abbott Diagnostics, USA), quantitative Spike-1/Spike-2 (S1/S2) IgG on LIASON-155 XL (DiaSorin S.p.A, Italy), and Spike (S1) IgG immunoassay (EUROIMMUN, Germany) were performed. Samples were reported positive if the signal was greater than the published cutoff value (>1.4). Signal to cutoff ratios were used.

## SARS-CoV-2 viral–cell fusion assay

The hACE2 ORF (Addgene# 1786) was cloned into a third-generation lentiviral expression vector, and clonal stable ACE2-expressing Hek293T cells were generated by lentiviral transductions [47]. Lentiviral particles pseudotyped with SARS-CoV-2 Spike Envelope were produced by co-transfecting Hek293T cells with a GFP-encoding lentiviral plasmid HRSIN-CSGW [48], psPAX2, and plasmid expressing carboxyl-terminal truncated Spike (pCG1-SARS-2-S Delta18) [49] including D614, D614G, S477N/D614G, B.1.1.7, and B.1.351 [47]. Neutralization activity of sera was measured using a single round infection of ACE2-HEK293T with Spike-pseudotyped lentiviral particles. Virus particles were incubated with serially diluted donor sera for 1 hour at 37°C. Virus–serum mix was then added onto ACE2-HEK293T cells ($2.5 \times 10^3$/well) in a 384-well plate. Following spinoculation at 1,200g for 1 hour at 18°C, the cells were moved to 37°C for 72 hours. Entry of Spike particles was imaged by GFP-positive cells (InCell Analyzer) followed by enumeration with InCarta software (Cytiva, USA). Uninfected cells were run in parallel to account for background fluorescence signal. In infected cells, Spike-driven entry of lentiviral particles resulted in strong GFP expression, while the uninfected controls were negative for GFP (S8A and S8B Fig). Neutralization was measured by reduction in GFP expression relative to control group infected with the virus particles without any serum treatment. Serum dilution resulting in 50% reduction in GFP expression relative to the infected control was used to determine virus–cell fusion inhibition titers (S8B and S8C Fig). The cutoff of positivity for virus–cell fusion inhibition was 1:40. Accordingly, fusion inhibition was considered as absent when titers were below 1:40. Pre-pandemic healthy control sera (Privigen USA, P100103528, 2091200221, P100007516) showed no fusion inhibition and hence no reduction in GFP expression, thus confirmimg the specificity of the assay (S8C Fig). Of the 10 NIBSC standards (CS678 WHO protocol), 5 were positive in the assay with mid and high virus–cell fusion inhibition, consistent with the confirmed results from the NIBSC.

The virus entry pathway in VeroE6, used in live-virus neutralization assays, is primarily endosomal [50]. In contrast, cells derived from nasopharyngeal tissues express ACE2 in addition to the surface serine protease TMPRSS2, which drives virus–cell membrane fusion and can signficantly enhance viral entry [49]. To address viral neutralization in the presence of ACE2 and TMPRSS2, a portfolio of Hek293T expressing clonal cell lines with ACE2 and TMPRSS2 (HekAT) was generated. The coexpression of ACE2 and TMPRSS2 led to a series of increasingly permissive cell lines that were readily susceptible to SARS-CoV-2 cytopathic effects, VeroE6, HekAT14, HekAT10, and HekAT24 (S3 Fig).

## High-content fluorescent live SARS-CoV-2 neutralization assay

Sera were serially diluted and mixed in duplicate with an equal volume of virus solution at $1.5 \times 10^3$ TCID50/mL. After 1 hour of virus–serum coincubation at 37˚C, 40 µL were added to equal volume of freshly trypsinized VeroE6 cells, and 3 clonal HekAT cells selected on SARS-CoV-2 permissiveness in 384-well plates ($5 \times 10^3$/well). After 72 hours, cells were stained with NucBlue (Invitrogen, USA), and each entire well was imaged with InCell Analyzer. Nuclei counts, proxy for cytopathic effect, were compared between convalescent sera, mock controls (defined as 100% neutralization), and infected controls (defined as 0% neutralization) using the formula; % viral neutralization = (D-(1-Q))x100/D, where Q = nuclei count normalized to mock controls and D = 1-Q for average of infection controls (InCarta software) (S8D and S8E Fig). The cutoff for determining the neutralization endpoint titer of diluted serum samples was set to ≥50% neutralization (S8F Fig), and the cutoff of positivity for neutralization on live virus was 1:40 [51]. During assay validation, 5 out of the 10 NIBSC standards (CS678 WHO protocol) were positive in the assay with mid and high levels of live neutralization, consistent with the confirmed results from the NIBSC.

## Enumeration of SARS-CoV-2 particles

Live SARS-CoV-2 and lentiviral particles were stained using SYTO RNASelect Green Fluorescent cell Stain (Invitrogen) at a final concentration of 10 µM for 30 minutes at 37˚C in freshly thawed unpurified viral particles. Particles were then diluted 1/10 and 1/100 in sterile PBS and then adhered to Poly-L-Lysine coated glass bottom 96-well Greiner Sensoplates (Sigma-Aldrich, USA) through spinoculation at 1,200 g for 1 hour at 18˚C. Particles were either imaged live or immune-fluorescently counterstained using a rabbit polyclonal SARS-CoV-2 Nucleocapsid antibody, followed by Alexa647-conjugated goat anti-rabbit IgG (Novus Biologicals, USA). Viral particles were then imaged and quantified as previously described [52]. Particle to infectivity ratios were determined by dividing the total particle count per ml with the calculated TCID50/ml. Particle to GFP transduction ratios were used for lentiviruses.

## SARS-CoV-2 Spike sequencing and analysis

Clinical respiratory samples were sequenced using an existing amplicon-based Illumina sequencing approach. The raw sequence data were subjected to an in-house quality control procedure before further analysis as reported in [53]. Non-synonymous SARS-CoV-2 Spike mutations (read frequency >0.8, minimum coverage 10×) were inferred from variant calling files during bioinformatic analysis using phylogenetic assignment of named global outbreak lineages (PANGOLIN) [11]. All consensus SARS-CoV-2 genomes identified have been uploaded to GISAID (www.gisaid.org).

## S477N glycosylation analysis

DNA encoding SARS-Cov-2 RBD (D614G variant; residues 319 to 541) was gene synthesized (Genscript) and cloned into pCEP4 mammalian expression vector with an N-terminal IgG leader sequence and carboxyl-terminal Avitag and His tag. Overlap extension polymerase chain reaction (OE-PCR) using primers harboring the S477N mutation was performed to generate the mutant plasmid. The plasmid was transfected into Expi293 cells (Thermo Fisher Scientific) according to the manufacturer's protocol, and the protein expressed for 7 days at 37˚C, 5% $CO_2$. The cell culture was clarified by centrifugation, dialyzed with PBS and the protein captured with Talon resin. The RBD was eluted with 150 mM imidazole in PBS and dialyzed against PBS. Purified RBD at 250 µg/mL was incubated with 500 units of PNGase F (NEB) at

37 °C for 1 hour under denaturing reaction conditions and analyzed by SDS-PAGE gel electrophoresis. The Fc region of human IgG1 (carrying a single glycosylation site at position 297) was used as a control.

## Statistics

Statistical analyses were performed in R v4.0.3. Loess curves were generated using ggplot2 v3.3.2. For categorical variables, a log-linear model was fitted and Pearson residuals plotted in a mosaic plot (MASS v7.3–51.6). Shapiro–Wilk test was used to test for normality in continuous variables, and Dwass–Steel–Critchlow–Fligner and Mann–Whittney tests were used to test for significance between continuous and categorical variables. Comparisons between clinical groups (Fig 3) were performed using general linear models (GLMs) (gaussian) to model the response (MFI) and explanatory variables, such as sex and age. ANOVA likelihood ratio test was used to test whether interactions need be included in the final model. Adjusted $P$ values were calculated using Tukey honestly significant difference (HSD) test. Correlations were measured using the Spearman method (psych v2.0.8). Virus–cell fusion and neutralization data were fitted using an exponential decay curve (OriginLab, USA). Patient curves unable to be fitted, $<3$ collection dates, or low viral fusion and neutralization were undetermined. Statistical significance was determined as $P < 0.05$.

## Supporting information

**S1 Fig. Correlation matrices for antibody responses and demographic data in ADAPT (A) and LIFE (B).** Continuous measures of demographics (age, days post-PCR positivity) and antibody titers (Nucleocapsid, Spike IgG and IgM, and Membrane), were compared. R-square values are placed in the boxes (top-right) with $P$ value significance shown in $*$. Bottom-left show correlation plots with Loess line (red). PCR, polymerase chain reaction.
(TIF)

**S2 Fig. Spike IgG decay profile of high plasmapheresis donors.** Spike IgG titer over time is shown from $n = 30$ LIFE donors who underwent $>10$ plasmapheresis donations. Despite high donations, donors with high Spike IgG titers decreased but stabilized at mid-levels, whereas donors with mid to low Spike IgG titers stabilized at low titers. No donors seroreverted nor became negative for Spike IgG.
(TIF)

**S3 Fig. Hyperpermissiveness of Hek cell lines.** HekAT clonal cell lines and VeroE6 were infected with serially diluted SARS-CoV-2 and monitored for CPE at 72 hours postinfection. **(A)** Cell nuclei were stained with NucBlue and CPE quantified as % live cells. HekAT clones showed varying degrees of permissiveness to SARS-CoV-2 infection with HekAT24 being orders of magnitude more susceptible than VeroE6. WT Hek293T cells were refractory to infection and were used as negative control. **(B)** Bright-field images showing CPE in WT Hek and HekAT clonal lines. Images were acquired using InCell high throughput imaging system. Magnification is 10× for all images. Representative images are shown. CPE, cytopathic effect; SARS-CoV-2, Severe Acute Respiratory Syndrome Coronavirus 2; WT, wild-type.
(TIF)

**S4 Fig. GFP expression and positive individual status against 2 new Spike variants. (A** and **B)** Expression of GFP reporter molecule expressed via the transcription of Spike-GFP monocistron in the pIRES2 plasmids was similar across all analyzed variants, allowing accurate comparisons between IgG levels between variants. (**B** and **C**) Pre-pandemic controls were below the threshold, whereas all first and second wave of ADAPT samples were above the positive

threshold (Control + 4SD) in (**B**) all D614G Spike variants and in (**C**) VOC B.1.1.7 (United Kingdom) and B.1.351 (South African). VOC, variant of concern.
(TIF)

**S5 Fig. Individuals with reduced immunoreactivity to D614G Spike variant were more likely to be females.** Mosaic plots of ADAPT (left) and LIFE (right) show individuals with restricted binding to Spike variant, i.e., reduced binding to D614G, were more likely to be females.
(TIF)

**S6 Fig. S477N Spike variant was not N-glycosylated.** SDS-PAGE of purified Wuhan-1 (WT) and S477N RBD expressed in human Expi293 cells under reducing conditions (stained with Coomassie blue). N-linked carbohydrates were removed by treatment with PNGaseF (+). No additional N gly-cosylation could be observed in S477N compared to Wuhan-1 (WT). The Fc region of human IgG1 (carrying a single glycosylation site at position 297) was used as a control. WT, wild-type.
(TIF)

**S7 Fig. SARS-CoV-2 antibody responses in high and elite responders maintain neutraliza-tion to emerging Spike variants.** (I) High and elite sera maintained high titers of neutraliza-tion to live VOC B.1.1.7, B1.351, B1.1.28.1, and B1.1.28.2. SARS-CoV-2, Severe Acute Respiratory Syndrome Coronavirus 2; VOC, variant of concern.
(TIF)

**S8 Fig. SARS-CoV-2 virus–cell fusion and high-content fluorescent live SARS-CoV-2 neu-tralization assay. (A)** Uninfected and infected ACE2-Hek 293T cells in Spike-driven virus–cell fusion assay. Spike-mediated entry of lentiviral/pseudoviral particles resulted in robust GFP expression in infected cells, while the uninfected cells showed no GFP signal (top). Cell nuclei were stained with NucBlue (bottom). **(B)** Mean intensity values of GFP signal in unin-fected and infected control cells. Data points represent pooled technical replicates. (**C**) Serum titration curves for a high neutralizing convalescent serum and sera from pre-pandemic healthy controls. Dilution resulting in 50% reduction in GFP expression relative to the infected control was taken as the cutoff to determine fusion inhibition titers. **(D)** Uninfected and infected VeroE6 cells in SARS-CoV-2 live neutralization assay. Cell nuclei were stained with NucBlue. Infection with live virus resulted in abundant cytopathic effect and cell death at 72 hours postinfection, leading to lower cell numbers compared to uninfected control. (**E**) The average uninfected nuclei counts (data points represented pooled technical replicates) was defined as 100% neutralization, whereas the average of infected nuclei counts was defined as 0% neutralization. The % viral neutralization of a sample was calculated using the formula described in Materials and methods. (**F**) Serum titration curves for a high neutralizing conva-lescent serum and sera from pre-pandemic healthy controls. The cutoff for determining the neutralization titer of diluted serum samples was ≥50%. Magnification is 10× for all images. Representative data from 2 experiments are shown. Mean and standard deviation are shown. SARS-CoV-2, Severe Acute Respiratory Syndrome Coronavirus 2.
(TIF)

**S1 Table. Nomenclature of SARS-CoV-2 and emerging variants.**
(PDF)

## Acknowledgments

We thank all the patients and donors who participated in this study. We thank Drs. Suat Der-vish, Edwin Lau, and Maggie Wang for providing advice at the Flow Cytometry Core Facility

of the Westmead Research Hub. We thank Ms. Rebecca Rielly, the Operation, and the Research Ethics and Governance teams at Kids Research for providing acces to the PC2 facility and advice on ethics and governance matters.

## Author Contributions

**Conceptualization:** Fiona Tea, Anthony D. Kelleher, Gail V. Matthews, Fabienne Brilot, Stuart G. Turville.

**Data curation:** Fiona Tea, Anupriya Aggarwal, David Ross Darley, Deepti Pilli, Daniele Vitale, Veronica C. Hoad, David O. Irving, Iain B. Gosbell, Gail V. Matthews.

**Formal analysis:** Fiona Tea, Alberto Ospina Stella, Anupriya Aggarwal, David Ross Darley, Deepti Pilli, Daniele Vitale, Philip Cunningham, Ohan Mazigi, Daniel Christ, Rebecca J. Rockett, Vitali Sintchenko, Anthony D. Kelleher, Gail V. Matthews, Fabienne Brilot, Stuart G. Turville.

**Funding acquisition:** Anthony D. Kelleher, Fabienne Brilot, Stuart G. Turville.

**Investigation:** Fiona Tea, David Ross Darley, Deepti Pilli, Daniele Vitale, Vera Merheb, Fiona X. Z. Lee, Philip Cunningham, Gregory J. Walker, William D. Rawlinson, Vennila Mathivanan, Markus Hoffmann, Stefan Pöhlman, Dominic E. Dwyer, Rebecca J. Rockett, Veronica C. Hoad, Iain B. Gosbell, Anthony D. Kelleher, Stuart G. Turville.

**Methodology:** Fiona Tea, Alberto Ospina Stella, Anupriya Aggarwal, Deepti Pilli, Daniele Vitale, Vera Merheb, Fiona X. Z. Lee, Gregory J. Walker, Christina Fichter, David A. Brown, William D. Rawlinson, Sonia R. Isaacs, Vennila Mathivanan, Markus Hoffmann, Stefan Pöhlman, Ohan Mazigi, Daniel Christ, Rebecca J. Rockett, Vitali Sintchenko, David O. Irving, Gregory J. Dore, Fabienne Brilot.

**Project administration:** Fabienne Brilot.

**Resources:** Gregory J. Dore, Iain B. Gosbell, Gail V. Matthews.

**Supervision:** Fabienne Brilot, Stuart G. Turville.

**Writing – original draft:** Fiona Tea, Alberto Ospina Stella, Anupriya Aggarwal, David Ross Darley.

**Writing – review & editing:** Deepti Pilli, Daniele Vitale, Vera Merheb, Fiona X. Z. Lee, Philip Cunningham, Gregory J. Walker, Christina Fichter, David A. Brown, William D. Rawlinson, Sonia R. Isaacs, Vennila Mathivanan, Markus Hoffmann, Stefan Pöhlman, Ohan Mazigi, Daniel Christ, Dominic E. Dwyer, Rebecca J. Rockett, Vitali Sintchenko, Veronica C. Hoad, David O. Irving, Gregory J. Dore, Iain B. Gosbell, Anthony D. Kelleher, Gail V. Matthews, Fabienne Brilot, Stuart G. Turville.

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
