## [Editor Report · Decision Letter 0]

8 Feb 2021

Dear Dr Brilot, 

Thank you for submitting your manuscript entitled "SARS-CoV-2 neutralizing antibodies; longevity, breadth, and evasion by emerging viral variants" for consideration by PLOS Medicine.

Your manuscript has now been evaluated by the PLOS Medicine editorial staff as well as by an academic editor with relevant expertise and I am writing to let you know that we would like to send your submission out for external peer review.

Kind regards,

Dr Raffaella Bosurgi

Executive Editor 

PLOS Medicine

---

## [Decision Letter · Decision Letter 1]

22 Mar 2021

Dear Dr. Brilot,

Thank you very much for submitting your manuscript "SARS-CoV-2 neutralizing antibodies; longevity, breadth, and evasion by emerging viral variants" (PMEDICINE-D-21-00606R1) for consideration at PLOS Medicine. 

Your paper was evaluated by the editorial team (editors present Raffaella Bosurgi, Richard Turner, Caitlin Moyer, Beryne Odeny). It was sent to independent reviewers, including a statistical reviewer. The reviews are appended at the bottom of this email and any accompanying reviewer attachments can be seen via the link below:

[LINK]

In light of these reviews, I am afraid that we will not be able to accept the manuscript for publication in the journal in its current form, but we would like to consider a revised version that addresses the reviewers' and editors' comments. Obviously we cannot make any decision about publication until we have seen the revised manuscript and your response, and we plan to seek re-review by one or more of the reviewers. 

We expect to receive your revised manuscript by Apr 12 2021 11:59PM. Please email us (plosmedicine@plos.org) if you have any questions or concerns.

We look forward to receiving your revised manuscript. 

Sincerely,

Dr Raffaella Bosurgi, 

Executive Editor 

PLOS Medicine

plosmedicine.org

Comments from the reviewers:

Reviewer #1: This article titles "SARS-CoV-2 neutralizing antibodies; longevity, breadth, and evasion by emerging viral variants" benefits from a reasonably large cohort followed for up to seven months.

It presents a thorough and rigorous set of analyses, applying technically appropriate statistical methods.

The findings are highly topical and important, providing a valuable and timely platform for urgent further research.

Reviewer #3: Reviewer José Alcamí

In this article Tea et al. Gives a comprehensive and broad perspective on the evolution and role of humoral responses against SARS-CoV-2. To this aim they study two Autralian cohorts, an hospital-based cohort (ADAPT n=100 patients) and a cohort of plasma donors (LIFE, n=159). The article is interesting but some of the data have been already published by other authors, which limits the originality of the results. The study asses humoral responses with different techniques. Main results are the longevity of antibody responses, particularly IgM antibodies that were detectable later than in previous articles. A correlation between IgG levels and neutralization capacity is shown. A subgroup of patients displaying high, broad and persistent humoral responses were identified. These patients would be good donors of hyperimmune plasma and besides high IgG levels they display also IgM antibodies which is an interesting observation. However, neutralization against emerging variants were significantly reduced in a proportion of patients. It this context it is interesting the decreased neutralization against 614G variants from patients infected with the original Wuhan 614D strain, an observation that as far as I know has not been reported so clearly.

Major comments:

1. The authors determine SARS-CoV-2 antibodies against surface viral proteins (S, M and E) using a flow-cytometry assay. As the authors propose that this test should be used for assessment of antibody responses at it enhances the sensitivity for detection of Spike antibodies (Discussion lines 353-356) in comparison with conventional ELISA tests (Table 2, Euroimmun test and Diasorin) standard controls should be incorporated to rule out false positive results in the flow cytometry assay. 

2. From figure 1A and methods (lines 477-483) it is not clear if Membrane and/or Envelope proteins were expressed individually in different cell lines or in the same cell line. Please clarify.

3. Related with the last point, in figure 1D membrane IgG antibodies are quantified. Did the authors quantify anti-Envelope antibodies as part of polyantigenic breadth? (lines 155-167).

4. Emerging variants studies are interesting but their relevance is low taking into account the current circulating Variants of Concern.

Minor comments: 

1. First sample was obtained beyond two months after a positive PCR test. Thus, one limitation of the study is the assessment of early antibody responses anti-SARS-CoV-2

2. Relationship between higher immunoglobulin levels and age probably is due to more severe disease in elderly people. This point should be mentioned.

3. Articles that characterize the presence of memory B-lymphocytes targeting the Spike protein should be mentioned as a proof of long-lasting immune memory (Papers from Shane Crotty for example). 

In summary, the article describes in two cohorts of patients with different clinical severity the range and evolution of IgM and IgG antibodies against different SARS-CoV-2 proteins using several techniques that combine ELISA, flow-cytometry and neutralization testing. Results are consistent and both technical approaches and some data are relevant. My major criticism is the minor originality of some findings that have already been reported. To be published in Plos Medicine, to include the reactivity and neutralization capacity of the studied plasma against current variants of concern (B.1.1.7, B1.351 and B.1.1.248) would increase the interest of the article.

[LINK]

---

## [Editor Report · Decision Letter 2]

5 May 2021

Dear Dr. Brilot,

Thank you very much for re-submitting your manuscript "SARS-CoV-2 neutralizing antibodies; longevity, breadth, and evasion by emerging viral variants" (PMEDICINE-D-21-00606R2) for review by PLOS Medicine.

I have discussed the paper with my colleagues and the academic editor James Beeson I am pleased to say that provided the remaining editorial and AE's points and production issues are dealt with we are planning to accept the paper for publication in the journal.

AE points:

My main outstanding issue with the paper is the reporting of statistical significance. There are a few problems the authors need to address

1. There needs to be consistent citation of p values where they state a difference is significant. There are several places where they report significant differences, but no p value provided in the text, and often it is missing in the figure/legend. They also describe higher or lower antibodies in different contexts in the text of the paper without indicating whether the differences are significant

2. They need to either indicate the p value or significance (or NS) in the figures themselves, or include them in the figure legend. Since many of their figures are very small, adding p values to the figure legend would be better and clearer.

3. Many figures are lacking any indication of a p value, either in the figure, the legend, or in the manuscript text. My recommendation to make this more consistent and accessible to readers is for them to put p values for each figure in the legend. this needs to be done for all figure sub-parts. If the difference is not significant, they should state the exact p value

4. When comparing differences in antibodies between clinical groups, there is no mention if they adjusted for covariates, or how they did this.

[LINK]

We look forward to receiving the revised manuscript by May 12 2021 11:59PM.   

Sincerely,

Dr Raffaella Bosurgi, 

Executive Editor 

PLOS Medicine

plosmedicine.org

Requests from Editors:

Comments from Reviewers:

[LINK]

---

## [Editor Report · Decision Letter 3]

12 May 2021

Dear Dr Brilot, 

On behalf of my colleagues and the Academic Editor, James Beeson, I am pleased to inform you that we have agreed to publish your manuscript "SARS-CoV-2 neutralizing antibodies; longevity, breadth, and evasion by emerging viral variants" (PMEDICINE-D-21-00606R3) in PLOS Medicine.

PRESS

Sincerely, 

Dr Raffaella Bosurgi 

Executive Editor

PLOS Medicine